# Targeting MYC dependency in ovarian cancer through inhibition of CDK7 and CDK12/13

Mei Zeng[1,2], Nicholas P Kwiatkowski[1,2], Tinghu Zhang[1,2], Behnam Nabet[1,2], Mousheng Xu[1,3], Yanke Liang[1,2], Chunshan Quan[1,2], Jinhua Wang[1,2], Mingfeng Hao[1,2], Sangeetha Palakurthi[4], Shan Zhou[4], Qing Zeng[4], Paul T Kirschmeier[4], Khyati Meghani[5], Alan L Leggett[1,2], Jun Qi[1,3], Geoffrey I Shapiro[6], Joyce F Liu[7], Ursula A Matulonis[7], Charles Y Lin[8,9]*, Panagiotis A Konstantinopoulos[7]*, Nathanael S Gray[1,2]*

[1]Department of Cancer Biology, Dana-Farber Cancer Institute, Boston, United States; [2]Department of Biological Chemistry and Molecular Pharmacology, Harvard Medical School, Boston, United States; [3]Department of Medicine, Harvard Medical School, Boston, United States; [4]Belfer Center for Applied Cancer Science, Dana-Farber Cancer Institute, Boston, United States; [5]Department of Radiation Oncology, Dana-Farber Cancer Institute, Boston, United States; [6]Early Drug Development Center, Dana-Farber Cancer Institute, Boston, United States; [7]Department of Medical Oncology, Dana-Farber Cancer Institute, Boston, United States; [8]Department of Biochemistry and Molecular Biology, Baylor College of Medicine, Houston, United States; [9]Department of Molecular and Human Genetics, Baylor College of Medicine, Houston, United States

*For correspondence:
Charles.Y.Lin@bcm.edu (CYL);
Panagiotis_Konstantinopoulos@
DFCI.HARVARD.EDU (PAK);
nathanael_gray@dfci.harvard.edu
(NSG)

Competing interest: See
page 17

Reviewing editor: Ross L
Levine, Memorial Sloan-Kettering
Cancer Center, United States

**Abstract** High-grade serous ovarian cancer is characterized by extensive copy number alterations, among which the amplification of *MYC* oncogene occurs in nearly half of tumors. We demonstrate that ovarian cancer cells highly depend on MYC for maintaining their oncogenic growth, indicating MYC as a therapeutic target for this difficult-to-treat malignancy. However, targeting MYC directly has proven difficult. We screen small molecules targeting transcriptional and epigenetic regulation, and find that THZ1 - a chemical inhibiting CDK7, CDK12, and CDK13 - markedly downregulates MYC. Notably, abolishing MYC expression cannot be achieved by targeting CDK7 alone, but requires the combined inhibition of CDK7, CDK12, and CDK13. In 11 patient-derived xenografts models derived from heavily pre-treated ovarian cancer patients, administration of THZ1 induces significant tumor growth inhibition with concurrent abrogation of MYC expression. Our study indicates that targeting these transcriptional CDKs with agents such as THZ1 may be an effective approach for MYC-dependent ovarian malignancies.
DOI: https://doi.org/10.7554/eLife.39030.001

## Introduction

Epithelial ovarian cancer (OC) is the fifth most common cause of female cancer death in the United States and the most lethal gynecologic malignancy (*Siegel et al., 2017*). High-grade serous ovarian carcinoma (HGSOC) represents the most common and aggressive histologic subtype of OC, and accounts for the majority of its deaths (*Konstantinopoulos and Awtrey, 2012*). Large-scale genomic studies have demonstrated that HGSOCs are characterized by high degree of genomic instability with high frequency of DNA copy number alterations and almost universal presence of TP53

mutations (*Cancer Genome Atlas Research Network, 2011*). Approximately 50% of HGSOCs exhibit an underlying defect in DNA repair via homologous recombination (HR) and are highly sensitive to double-strand-break-inducing agents such as platinum analogues and PARP-inhibitors (PARPi) (*Konstantinopoulos et al., 2015*). However, although first-line platinum-based chemotherapy results in clinically complete remissions in approximately 70% of OC patients, relapse occurs in more than 90% of these patients, at which point the disease is much less responsive to subsequent treatment and is essentially non-curable. Similarly, despite initial responses to PARPi among HR-deficient HGSOCs, acquired resistance occurs commonly and represents a significant barrier to the long-term survival of these patients (*Lord and Ashworth, 2017*). Overall, the outlook for patients with platinum and PARPi-resistant disease is poor, so novel therapeutic strategies are urgently needed (*Vaughan et al., 2011*).

The control of gene transcription involves a set of cyclin-dependent kinases (CDKs), including CDK7, CDK8, CDK9, CDK11, CDK12, CDK13, and CDK19, that play essential roles in transcription initiation and elongation by phosphorylating RNA polymerase II (RNAPII) and other components of the transcription apparatus (*Larochelle et al., 2007*; *Zhou et al., 2012*). Recent studies have shown that certain oncogenes, for example *MYC*, *MYCN*, and *RUNX1* exhibit significant dependence on continuous active transcription, and that inhibition of the general transcriptional machinery may allow for highly selective effects on these oncogenes in cancer cells before global downregulation of transcription occurs (*Kwiatkowski et al., 2014*; *Cao and Shilatifard, 2014*; *Chipumuro et al., 2014*). The continuous active transcription of these oncogenes in cancer cells is often driven by exceptionally large clustered enhancer regions, termed super-enhancers, which are densely occupied by transcription factors and co-factors (*Hnisz et al., 2013*; *Lovén et al., 2013*). In this vein, it was recently shown that CDK7 mediates transcriptional addiction to a vital cluster of genes associated with super-enhancers in triple-negative breast cancer (TNBC), and that TNBC cells are exceptionally dependent on CDK7 (*Wang et al., 2015*). The CDK7 covalent inhibitor THZ1, which also inhibits the closely related kinases CDK12 and CDK13 (CDK12/13), has been also shown to directly suppress super-enhancer-associated oncogenic transcription in T-cell acute lymphoblastic leukemia, neuroblastoma and small cell lung cancer (*Kwiatkowski et al., 2014*; *Chipumuro et al., 2014*; *Christensen et al., 2014*).

Here, we identified THZ1 as a highly potent compound that downregulates MYC expression. THZ1 demonstrates exceptional in vivo activity in patient-derived xenograft (PDX) models of ovarian cancer that were platinum and PARPi resistant. Notably, suppression of MYC was only achieved by simultaneous inhibition of CDK7, CDK12, and CDK13. Our data suggest that combined inhibition of transcriptional CDKs with THZ1, or its derivatives, may be an effective approach for treating MYC-dependent ovarian cancer.

## Results and discussion

### MYC is frequently amplified in ovarian cancer and is essential for cancer cell growth

Previous large-scale studies of HGSOC demonstrated extensive copy number alterations (*Cancer Genome Atlas Research Network, 2011*). Among the total eight recurrent chromosome-arm gains, chromosome 8q has the most significant gains and occurred in 65% of the tumors (n = 489) (*Cancer Genome Atlas Research Network, 2011*). Analyzing the updated TCGA dataset that includes more patient samples also indicate the widespread 8q gain, in addition to 8 p loss (*Figure 1A*).

Inspired by earlier investigations of ovarian cancer reporting the amplification of 8q regions as well as that of *MYC* oncogene in 8q24 (*Baker et al., 1990*; *Etemadmoghadam et al., 2009*; *Staebler et al., 2006*), we focus on the amplification of *MYC* in ovarian cancer. Notably, ovarian cancer demonstrates the highest frequency of *MYC* amplification (*Figure 1B*), compared to many other tumor types. We further analyzed and found a significant correlation between the gene copy number of *MYC* and its gene expression level (assayed by RNA sequencing) (*Figure 1C*). Therefore, *MYC* has widespread amplification in ovarian cancer, and its amplification typically correlates with high-level expression of the *MYC* oncogene.

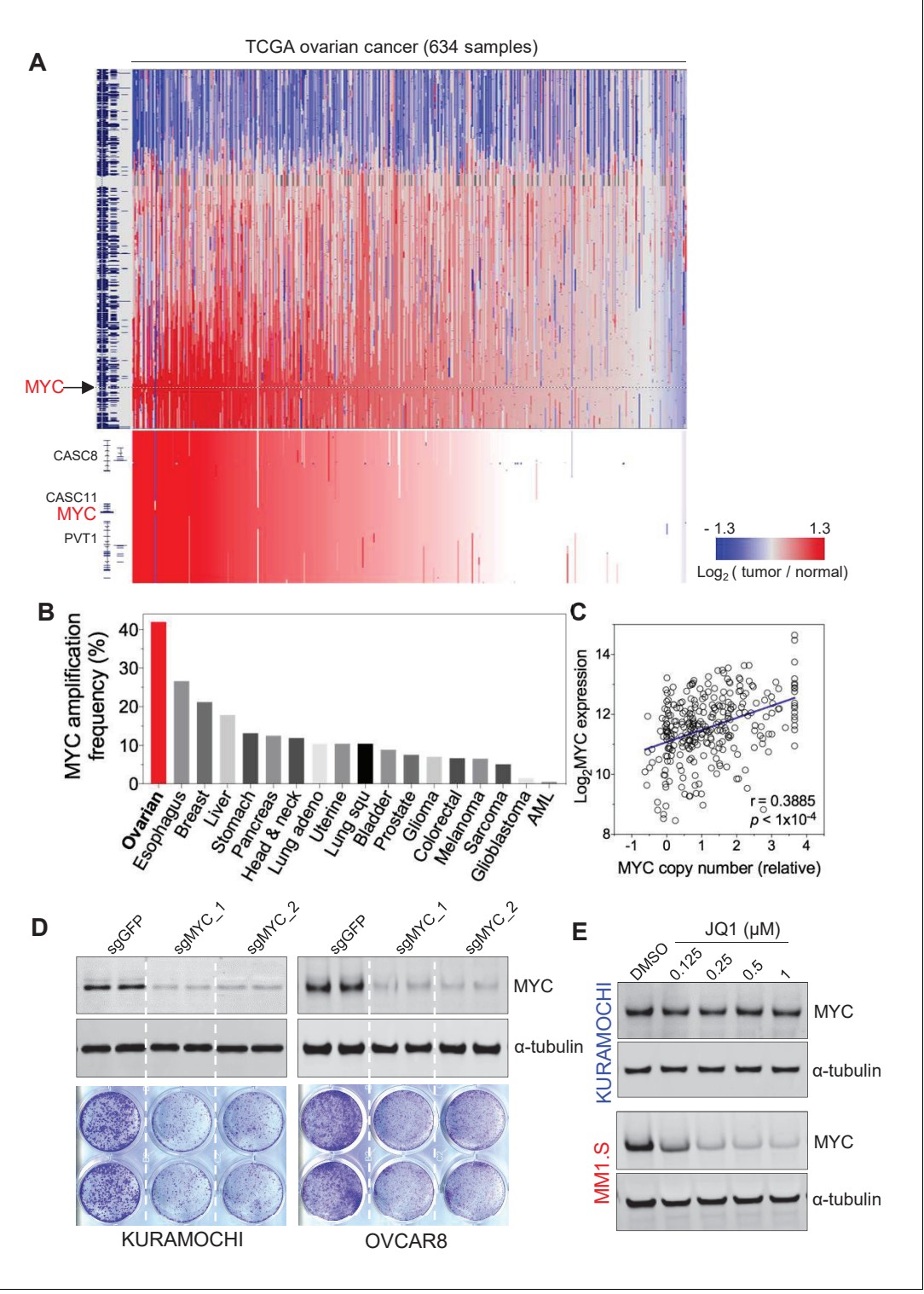

**Figure 1.** *MYC* is frequently amplified in ovarian cancer and required for cancer cell growth. (**A**) Copy number plots of TCGA high-grade serous ovarian cancer samples for chromosome 8 (top) and part of the q24 arm (bottom). Red color indicates a high chromosomal copy number ratio, blue represents low (see color key on the right). Data were analyzed and plotted using UCSC Xena Functional Genomics Browser (xena.ucsc.edu). (**B**) Frequency of *MYC* amplification across cancer types. (**C**) Correlation between *MYC* copy number and its gene expression in ovarian cancer. The relative copy number value and normalized RNA-seq expression values of *MYC* were downloaded from cBioportal and plotted in GraphPad Prism. Pearson correlation coefficient was measured

*Figure 1 continued on next page*

*Figure 1 continued*

and the p-value<$1\times10^{-4}$. (D) CRISPR/Cas9-mediated gene editing in ovarian cancer cells. Immunoblotting of lysates from ovarian cancer cells that were infected with lentivirus encoding Cas9 and sgRNA targeting *GFP* or *MYC*, and then harvested 2 days after puromycin selection (top). Cells were fixed after 12 days and stained with crystal violet (bottom). (E) Effect of JQ1 in ovarian cancer cells (top) and in a multiple myeloma line MM1.S (bottom). Cells were treated with vehicle control (DMSO) or increasing concentrations of JQ1 for 6 hr before lysates were prepared for immunoblotting with the indicated antibodies. Also see *Figure 1—figure supplement 1*.

DOI: https://doi.org/10.7554/eLife.39030.002

The following figure supplement is available for figure 1:

**Figure supplement 1.** Analysis of MYC dependency in cancer cell lines.

DOI: https://doi.org/10.7554/eLife.39030.003

Driven by the extensive alterations on both *MYC* gene copy and expression scales, we next proceeded to evaluate the functional role of MYC in OC lines. We utilized CRISPR/Cas9-mediated gene editing technique to disrupt the expression of MYC (*Sanjana et al., 2014*), and observed an efficient loss of MYC protein in cells infected with lenti-virus encoding two independent *MYC*-targeting guide RNAs (*Figure 1D*, top). MYC-depleted KURAMOCHI and OVCAR8 cells showed a significant deficit in cell viability compared to control cells as determined by clonogenic growth survival assay (*Figure 1D*, bottom). In addition, we analyzed the large-scale CRISPR screen performed by Broad Institute, a study where they developed the CERES computational model to reduce the false-positive differential dependencies caused by multiple DNA breaks resulted from targeting amplified regions (*Meyers et al., 2017*). In the analysis illustrating MYC dependency in a total of 484 cancer cell lines, there is no statistical correlation between MYC dependency values and with *MYC* copy number (*Figure 1—figure supplement 1A*), indicating the successful computational elimination of effects introduced by targeting amplified *MYC*. Notably, ovarian cancer cells overall demonstrate a high dependence on MYC (indicated by the low CERES values), and as expected, MYC dependency is highly correlated with cell dependency on MAX – a partner of MYC for transcriptional regulation (*Figure 1—figure supplement 1B*). These data further confirm the functional roles of MYC for ovarian cancer cell proliferation and indicate MYC as a promising therapeutic target for ovarian cancer.

## Screening transcriptional/epigenetic regulators identifies THZ1 as a potent inhibitor for MYC expression in ovarian cancer cells

Given the prominent gene amplification/overexpression of MYC and its critical roles for ovarian cancer cell growth, we next explored pharmacologic strategies for targeting MYC. Since the MYC protein lacks characteristics enabling specific and direct binding to small molecule compounds, recent studies have focused on approaches to interrupt the genesis of MYC transcript/protein or key downstream functions of MYC. In this regard, several studies have shown that inhibition of BET bromodomain proteins can effectively downregulate MYC transcription and consequently the growth of MYC-dependent cancer cells (*Delmore et al., 2011*; *Mertz et al., 2011*). Consistent with previous studies (*Delmore et al., 2011*), we found that MYC expression in the multiple myeloma line MM1.S is highly sensitive to JQ1; treating cells with JQ1 at nanomolar concentrations was sufficient to downregulate MYC protein abundance (*Figure 1E*, bottom). Surprisingly, this effect was not observed in ovarian cancer cells (*Figure 1E*, top), a phenotype reminiscent of that seen in other non-hematologic cancer cells, such as lung adenocarcinoma cells (*Lockwood et al., 2012*), and some triple-negative breast cancer cells (*Shu et al., 2016*).

To identify regulatory pathways that control *MYC* gene transcription in ovarian cancer, we proceeded to screen a selected group of chemicals. We selected 42 compounds, derived from the HMS small molecule library (http://lincs.hms.harvard.edu/db/sm/), targeting various transcriptional and/or epigenetic components, such as histone modification enzymes, transcriptional CDKs, transcriptional co-activators, and DNA modification enzymes (*Figure 2A*; *Supplementary file 1*). The compounds were used for treating two OC cell lines, KURAMOCHI and COV362. Whole cell lysates were then harvested for fluorescent immunoblotting, and MYC protein signal was normalized to that of loading control in the same membrane (*Figure 2A*; *Figure 2—figure supplement 1*). In both cell lines, these inhibitors display various effects on MYC protein abundance, and the most potent inhibitor is THZ1,

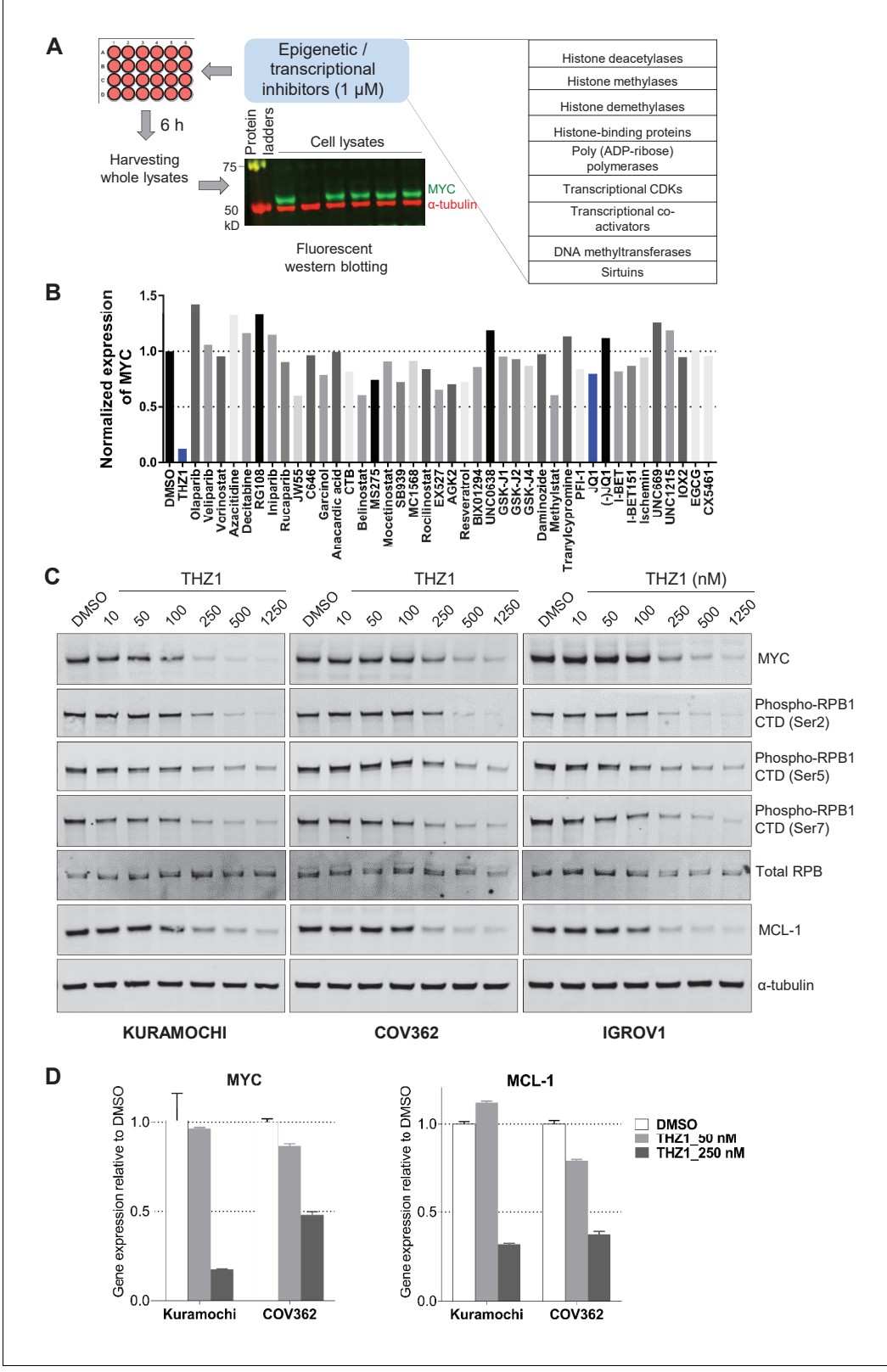

**Figure 2.** Targeted screen of transcriptional and epigenetic regulators identifies THZ1 as key transcriptional regulator of MYC.  (**A**) Schematic diagram of a screen for small molecules that inhibit MYC expression. Ovarian cancer cells were treated with selected epigenetic/transcriptional inhibitors for 6 hr at a final concentration of 1 µM. Whole cell lysates were subjected to fluorescent immunoblotting with anti-MYC and anti-α-tubulin (loading

*Figure 2 continued on next page*

*Figure 2 continued*

control) antibodies. (B) Normalized signals of MYC by immunoblotting in KURAMOCHI cells. THZ1 was the most potent inhibitor that reduces MYC expression. (C) The indicated cells were treated with increasing concentrations of THZ1 for 6 hr. Cell lysates were subjected to immunoblotting using the indicated antibodies. (D) qPCR analysis of *MYC* and *MCL-1* in THZ1-treated ovarian cancer cells. Student's t-test was performed and data were presented as mean values ± SD of technical triplicates. Also see *Figure 2—figure supplement 1*; *Figure 2—figure supplement 2*; *Supplementary file 1* and *Supplementary file 2*.

DOI: https://doi.org/10.7554/eLife.39030.004

The following figure supplements are available for figure 2:

**Figure supplement 1.** Small molecules screen for compounds inhibiting MYC expression.
DOI: https://doi.org/10.7554/eLife.39030.005
**Figure supplement 2.** THZ1 inhibits the expression of both MYC and MCL1.
DOI: https://doi.org/10.7554/eLife.39030.006

an inhibitor known to target CDKs 7, 12, and 13 (*Kwiatkowski et al., 2014*) (*Figure 2B*; *Figure 2—figure supplement 1*; *Supplementary file 2*).

We next validated the potency of THZ1 for inhibiting MYC expression in ovarian cancer cells. MYC protein abundance was decreased by THZ1 at ~250 nM and was further abolished when higher doses of THZ1 were used (*Figure 2C*). Consistent with the roles of CDK7 in regulating the phosphorylation of the carboxyl-terminal domain (CTD) of RNA polymerase II subunit RPB1, CTD phosphorylation at Ser2, 5, and 7 was suppressed by THZ1(*Figure 2C*).

## MCL-1 expression is sensitive to THZ1

Interestingly, the anti-apoptotic protein of Bcl-2 family, pro-survival protein myeloid cell leukemia 1 (MCL-1), was repressed by THZ1 concomitantly (*Figure 2C*; *Figure 2—figure supplement 1A–B*). We further examined the mRNA level of *MYC* and *MCL-1*, and found that treating both KURAMO-CHI and COV362 cells with THZ1 (250 nM) led to a 50% or greater reduction of both transcripts (*Figure 2D*), indicating that the transcriptional inhibition of *MYC* and *MCL-1* genes may be largely responsible for the observed reduction in protein abundance.

Although less prevalent than *MYC* amplification, *MCL-1* amplification was observed in 12% of HGSOCs of the TCGA dataset. Notably, in the TCGA, HGSOCs that harbored both *MYC* and *MCL-1* amplification were associated with poor outcome, which was significantly worse compared to all remaining HGSOCs (*Figure 2—figure supplement 2A*, left panel). Additionally, among *MYC* amplified HGSOCs, those that co-exhibited *MCL-1* amplification had significantly worse prognosis compared to *MYC* amplified tumors without *MCL-1* amplification (*Figure 2—figure supplement 2A*, right panel), indicating a role of MCL-1 during disease progression.

Interestingly, out of the total eight ovarian cancer cell lines (KURAMOCHI, COV362, OVCAR8, SKOV3, IGROV1, OVSAHO, OVCAR3, OVCAR4), the only cell line (SKOV3) that is least sensitive to THZ1 in terms of cell growth inhibition demonstrates low protein abundance of both MYC and MCL-1 (*Figure 2—figure supplement 2B–C*). While THZ1-sensitive lines, such as KURAMOCHI and OVACAR8, are susceptible to CRISPR/Cas9-mediated gene editing of *MYC* (*Figure 1D*) or *MCL-1* (*Figure 2—figure supplement 2D*), the THZ1-insensitive line SKOV3 was largely unaffected by the loss of MYC or MCL-1 (*Figure 2—figure supplement 2E*). These data suggest that the expression levels of MYC and MCL-1 in OC cells may determine its sensitivity to THZ1.

## THZ1 represses MYC target genes in ovarian cancer cells

We next performed global gene expression profiling in ovarian cancer cells to investigate the effects of THZ1 on transcriptional programs. In two THZ1-sensitive cell lines KURAMOCHI (Ku) and COV362 (Cov), we treated cells for 6 hr with vehicle control or THZ1 at two concentrations (50 and 250 nM) in duplicates, and then subjected samples for RNA isolation, library construction, and sequencing.

We analyzed the actively expressed genes (n = 14,068) in two ovarian cancer cell lines and the global change of gene expression was displayed in a concentration-dependent manner upon THZ1 treatment, shown by the heatmaps (*Figure 3A*). A profound suppression of gene expression was observed following treatment of THZ1 at 250 nM in both cell lines, with 1060 genes downregulated

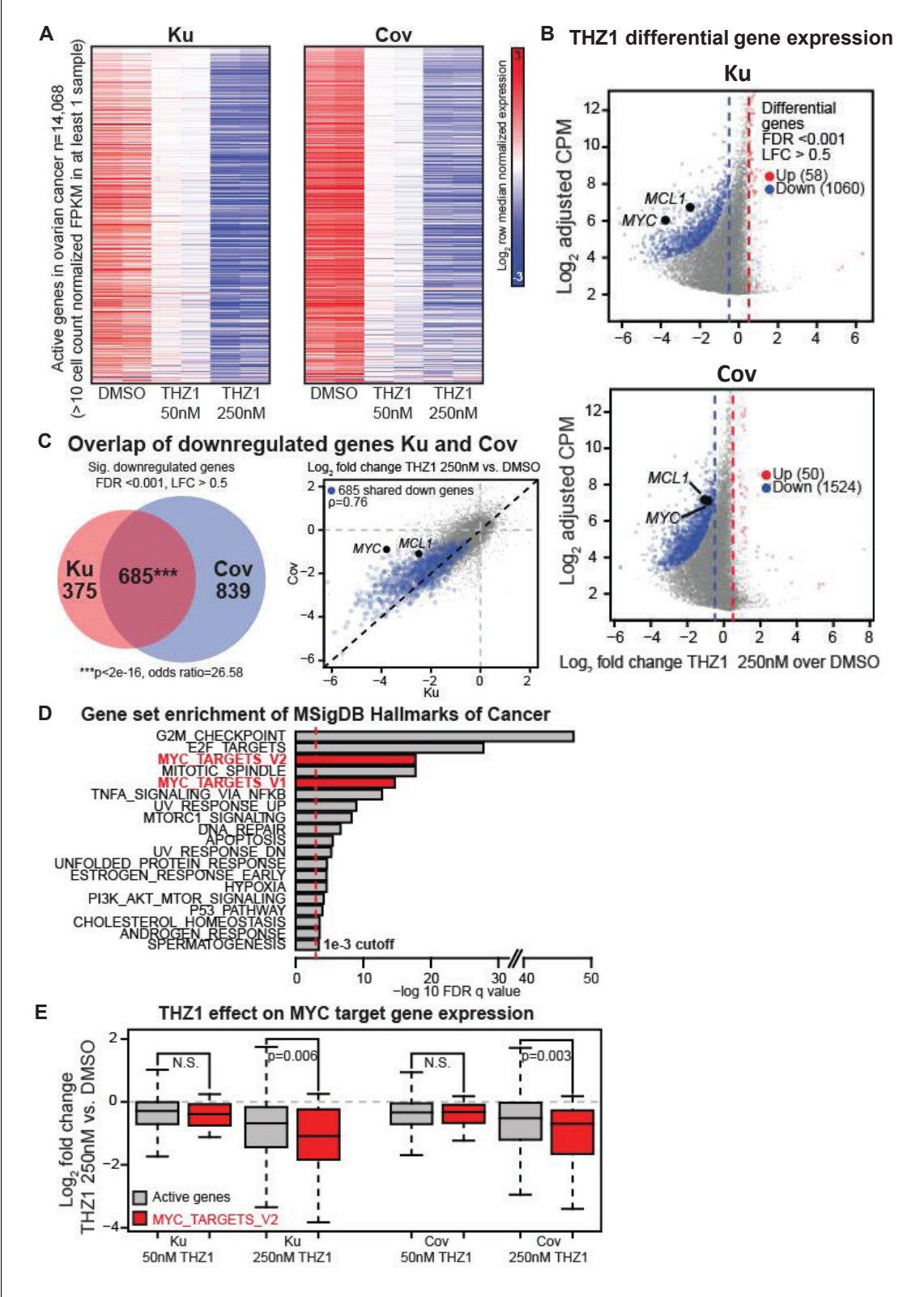

**Figure 3.** THZ1 represses MYC target genes in ovarian cancer cells. (**A**) Heatmaps showing row normalized gene expression for all actively expressed genes (n = 14,068) (cell count normalized FPKM > 1 in at least one sample). (**B**) Scatter plots comparing change in gene expression upon THZ1 treatment to gene expression levels in DMSO. X-axis shows the Log2 fold change of expression (THZ1/DMSO). Y-axis shows the adjusted log2 counts per million (CPM) expression in DMSO. Differentially regulated genes as determined by edgeR are shown for downregulated (blue) and upregulated

*Figure 3 continued on next page*

*Figure 3 continued*

(red) genes. A FDR adjusted p-value cutoff of 1e-3 and a log fold change (LFC) cutoff of 0.5 are used as the threshold for significance. (C) Left - Venn diagram showing the intersection of differential downregulated genes between Ku and Cov cells upon THZ1 250 nM treatment. Significance of overlap is determined by a fisher's exact test. p-value and odds ratio are shown. Right - Scatter plots comparing the log2 fold change in gene expression upon THZ1 250 nM treatment between Ku and Cov cells. Differential downregulated genes that are shared between Ku and Cov are shown in blue. The pearson correlation of differential genes is shown (Rho = 0.76). (D) Bar plot showing gene sets from the MSigDB Hallmarks of Cancer that are significantly enriched (FDR q value < 1e-3) amongst the 685 genes shared as differentially downregulated between Ku and Cov upon 250 nM THZ1 treatment. MYC target gene sets are shown in red. (E) Box plots showing the log2 fold change in gene expression upon THZ1 treatment for MYC target genes (red, n = 58) and other active genes (grey, n = 14,010) in Ku and Cov cell lines. MYC target genes are drawn from the HALLMARKS_MYC_TARGETS_V2 signature. The statistical significance between MYC target genes and other active genes is shown from a Wilcoxson rank sum test (one-sided). Also see *Figure 3—figure supplement 1*; *Figure 3—figure supplement 2*.

DOI: https://doi.org/10.7554/eLife.39030.007

The following figure supplements are available for figure 3:

**Figure supplement 1.** *PVT1*, a lncRNA gene co-amplified with *MYC* on 8q24, is not significantly downregulated by THZ1.

DOI: https://doi.org/10.7554/eLife.39030.008

**Figure supplement 2.** THZ1 selectively downregulates MYC target genes.

DOI: https://doi.org/10.7554/eLife.39030.009

and 58 genes upregulated in Ku cells (*Figure 3B*, top), and 1524 genes downregulated and 50 genes upregulated in Cov cells (*Figure 3B*), bottom). Among these, both *MYC* and *MCL-1* were significantly downregulated in both cell lines, with the cutoff of Log2 fold change >0.5 and FDR adjusted p-value<0.001 (*Figure 3B*). The regulation on *MYC* expression by THZ1 appears quite specific, because *PVT1* − a lncRNA gene co-amplified with *MYC* on 8q24 − is not significantly downregulated by THZ1 (*Figure 3—figure supplement 1*).

Next, we examined how many downregulated genes were shared between Ku and Cov cells and what the potential pathways these genes might be involved in. There were 685 differential downregulated genes been identified upon THZ1 250 nM treatment, shown by the Venn diagram (p-value for significance of overlap <2e-16) (*Figure 3C*, Left) and by the scatter plot(*Figure 3C*, Right), with the overlapped downregulated genes shown in blue (Pearson correlation coefficient Rho = 0.76). To search for the oncogenic pathways that were selectively downregulated by THZ1, we queried the Molecular Signature Database (MSigDB) Hallmarks of Cancer in Gene Set Enrichment Analysis (Broad Institute). G2M_checkpoint and E2F_targets appeared to be the top two hallmark gene sets that were significantly enriched amongst the 685 genes in Ku and Cov upon THZ1 250 nM treatment. Two MYC hallmark gene sets, MYC_targets_V1 and MYC_targets_V2 (shown in red), were also identified (*Figure 3D*). Additionally, these two MYC hallmark gene sets were also significantly enriched when analyzed individually (*Figure 3—figure supplement 2*). When comparing the MYC target genes (defined from the HALLMARKS_MYC_TARGETS_V2 gene set, in red) with all other active genes (in gray), they were significantly downregulated upon THZ1 250 nM treatment in both cell lines (*Figure 3E*). This analysis is consistent with THZ1 downregulating a MYC-dependent gene expression program.

## MYC and MCL-1 downregulation require the inhibition of both CDK7 and CDK12/13

Given the polypharmacology of THZ1, we asked whether the loss of MYC and MCL-1 resulted from inhibition of CDK7, or CDK12/13, or combined inhibition of CDK7 and CDK12/13. To do this, we turned to two recently described compounds that were selective for CDK7 (YKL-1–116, *Kalan et al., 2017*) or CDK12/13 (THZ531, *Zhang et al., 2016*) (*Figure 4A*). Compared with THZ1 (*Figure 2C*), YKL-1–116 exhibited similar but less potent effects on MYC downregulation in two OC cell lines (*Figure 4B–C*, *Figure 4—figure supplement 1A–B*). Different from THZ1 and YKL-1–116, the CDK12/13 inhibitor, THZ531, has a unique effect on MYC expression: at low doses of THZ531 MYC expression is induced, while higher doses of THZ531 cause MYC to be repressed (*Figure 4B–C*, *Figure 4—figure supplement 1A–B*). To test whether the combined CDK7 and CDK12/13 inhibition enhances MYC and MCL-1 downregulation, we treated cells with the combination of YKL-1–116 and THZ531. Interestingly, we found that the combination of YKL-1–116 and THZ531 led to pronounced reduction of MYC and MCL-1, recapitulating the effects of THZ1 (*Figure 4D–E*, *Figure 4—figure*

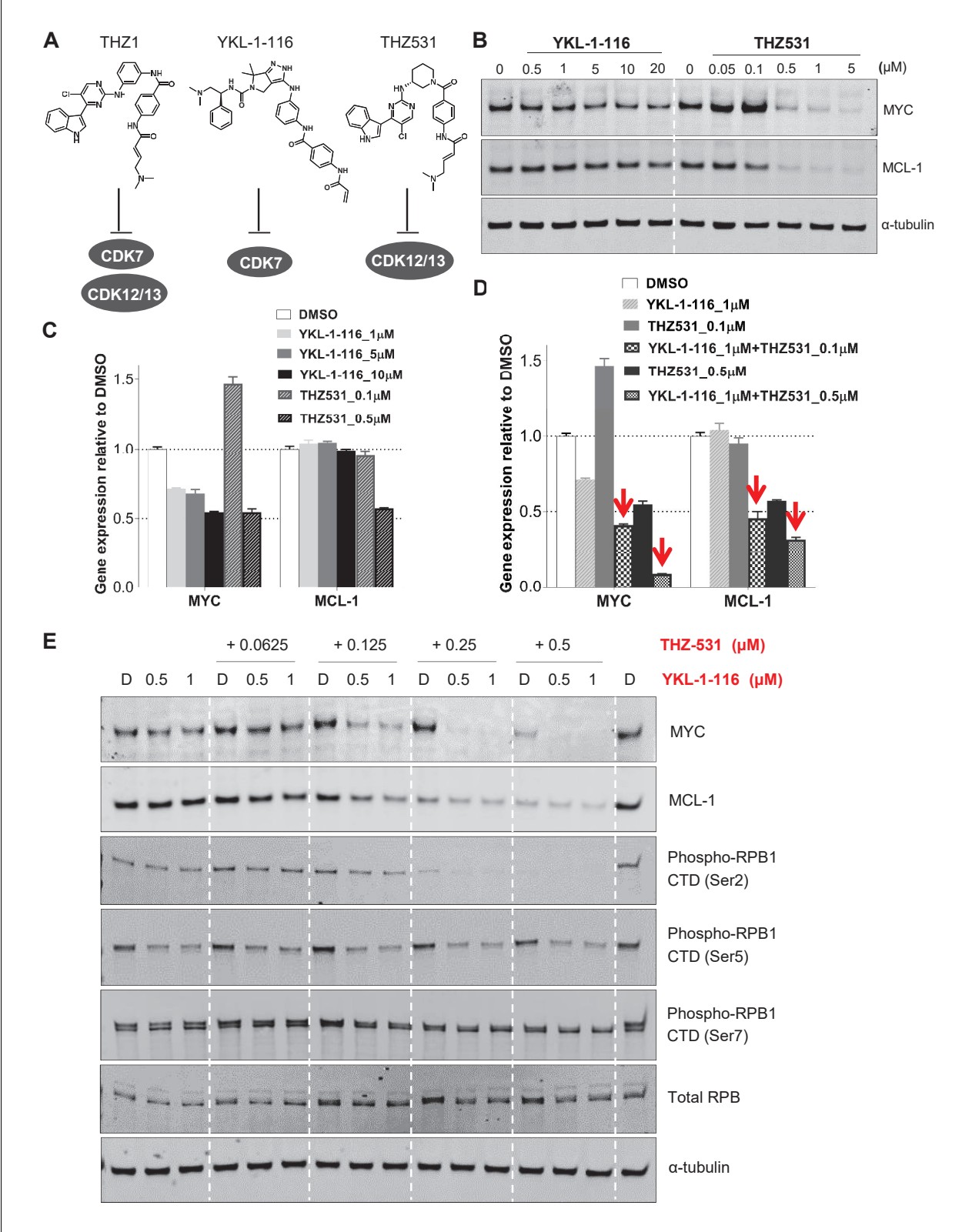

**Figure 4.** Downregulation of MYC and MCL-1 requires co-inhibition of CDK7 and CDK12/13 in KURAMOCHI cells. (A) Chemical structures of inhibitors and their primary protein targets. (B) KURAMOCHI cells were treated with increasing concentrations of YKL-1–116 or THZ531, with cell lysates prepared following 6 hr of treatment. (C) qPCR analysis of *MYC* and *MCL-1* in YKL-1–116 or THZ531-treated KURAMOCHI cells for 6 hr. Student's t-test was performed and data were presented as mean values ± SD of technical triplicates. (D) qPCR analysis of *MYC* and *MCL-1* in single or combination

*Figure 4 continued on next page*

*Figure 4 continued*

treatment with YKL-1–116 and THZ531 for 6 hr. Red arrows indicate an enhanced repression by the combination treatment. Student's t-test was performed and data were presented as mean values ± SD of technical triplicates.(E) Combining YKL-1–116 and THZ531 efficiently downregulates MYC expression. KURAMOCHI cells were treated with increasing concentrations of YKL-1–116 (0.5 or 1 μM), in combination with increasing concentrations of THZ531 (0.0625, 0.125, 0.25, or 0.5 μM) for 6 hr before lysates were prepared for immunoblotting. Also see *Figure 4—figure supplement 1*; *Figure 4—figure supplement 2*.

DOI: https://doi.org/10.7554/eLife.39030.010

The following figure supplements are available for figure 4:

**Figure supplement 1.** Downregulation of MYC and MCL-1 requires co-inhibition of CDK7 and CDK12/13 in COV362 cells.

DOI: https://doi.org/10.7554/eLife.39030.011

**Figure supplement 2.** The mutant CDK7 (C312S) rescued THZ1-induced MYC downregulation and cell growth inhibition by YKL-1–116.

DOI: https://doi.org/10.7554/eLife.39030.012

*supplement 1C–D*). In addition, the combination of YKL-1–116 and THZ531 further reduced the phosphorylation of RNA polymerase II RPB1 at Ser5, but not at Ser 2 and 7 (*Figure 4E* and *Figure 4—figure supplement 1D*). Although it remains to be understood how MYC and MCL-1 transcription requires the activity of both CDK7 and CDK12/13, our data indicate that the ability of THZ1 to downregulate MYC and MCL-1 is likely due to its multi-targeting effect on CDK7 and CDK12/13.

We next proceeded to investigate whether downregulation of MYC by THZ1 was mediated through CDK7 inhibition, we overexpressed HA-tagged wild-type CDK7 (WT) or C312S mutant (CS) in ovarian cancer cells by lentiviral infection. Cysteine 312 is the site for covalent modification, and its mutation to serine prevented THZ1 from covalently binding to CDK7 and from inhibiting CDK7 activity in an irreversible manner (*Kwiatkowski et al., 2014*). First, we examined whether downregulation of MYC by THZ1 can be rescued by CDK7 the C312S mutant. Expectedly, we find that the mutant CDK7 (C312S), but not the wild type, effectively rescues THZ1-induced MYC downregulation upon THZ1 treatment at 250 and 500 nM for 6 hr (*Figure 4—figure supplement 2A*). Concurrently, other readouts of CDK7 activity, such as CTD phosphorylation of RNAPII at Ser 5 and Ser 7, are also rescued by this CDK7 mutant (*Figure 4—figure supplement 2A*). These data indicate that CDK7 is indeed a target of THZ1 in downregulating MYC. Although overexpression of mutant CDK7 (C312S) fails to rescue cell growth inhibition conferred by THZ1 (*Figure 4—figure supplement 2B*), it significantly rescues cell growth inhibition by the selective CDK7 inhibitor, YKL-1–116 (*Figure 4—figure supplement 2B–2C*). We would expect that a greater degree of rescue might require the elimination of endogenous CDK7. In addition, considering that THZ1 targets both CDK7 and CDK12/13, we speculate that co-overexpression of mutant CDK7 (C312S), mutant CDK12 (C1039S), and mutant CDK13 may rescue cell growth inhibition by THZ1. Unfortunately, we failed to achieve overexpression of CDK12/13, likely due to their large sizes (1490 aa for CDK12, and 1512 aa for CDK13). Notably, expression of a THZ1-resistant mutant of CDK7 rescued THZ1-induced MYC downregulation but was not sufficient to rescue the aberrant cancer cell growth. This indicates that parallel pathways to MYC may also be altered by THZ1 and may contribute to the observed phenotypes upon THZ1 treatment.

## THZ1 suppresses the growth of patient-derived ovarian tumors

Next, we evaluated the therapeutic efficacy of THZ1 in ovarian tumor models. Although THZ1 targets expression of more genes than *MYC* via CDK7/12/13 inhibition, its high efficiency in downregulating *MYC* and other genes that are critical for ovarian cancer cells, such as *MCL-1*, provides strong rationale for translational development. We used the orthotopic ovarian patient-derived xenografts (PDX) models that we established previously (*Liu et al., 2017*). The primary tumor cells were transduced with luciferase gene to enable the use of bioluminescent imaging for measurement of tumor growth.

To compare the single agent potency of THZ1 with combination effect of drugs, we performed a combination study with THZ1 and PARP inhibitor Olaparib, a FDA-approved drug in relapsed ovarian cancer irrespective of BRCA1/2 status. We first conducted tolerability studies and found that THZ1 administered by intraperitoneal injection (IP) twice daily (BID) at 10 mg/kg was well-tolerated with no signs of overt toxicity as judged by body weight and animal behavior (data not shown). In efficacy studies, we first implanted ascites-derived ovarian tumor cells into the mice, and after 7 days

assigned animals into four groups receiving vehicle control (10 ml/kg, PO, QD) or THZ1 (10 mg/kg, IP, BID) or Olaparib (100 mg/kg, PO, QD) or combo (THZ1 +Olaparib) for 27 days, with bioluminescent imaging performed at 5 timepoints (0, 6, 13, 20, and 27 days) (*Figure 5A–B*). Consistent with previous studies of THZ1 or Olaparib, mouse body weight was minimally affected by the inhibitor (*Figure 5—figure supplement 1*). In all the 11 independent PDX models investigated, the administration of THZ1 caused significant inhibition on tumor cell growth (*Figure 5B–C*). Notably, in four models (DF-149, 172, 83, and 86), THZ1 induced complete inhibition on tumor growth (*Figure 4C*, termed category i). In six models (DF-101, 106, 118, 20, 68, and 216), THZ1 first caused an obvious decrease of tumor burden but re-gained growth at later time points (termed category ii). Only one model (DF-181, termed category iii) did not demonstrate tumor regression and rather present slower tumor cell growth upon THZ1 treatment. The administration of Olaparib did not dramatically inhibit tumor growth, and only showed very modest effect in three models (DF-106, 68, and 83). The combination of THZ1 and Olaparib, however, displayed synergistic effect and further inhibition on tumor growth was observed in five models (DF-106, 118, 86, 181, and 68).In addition, we found that the protein abundance of both MYC and MCL-1 in the tumor was nearly abrogated following THZ1 treatment (*Figure 5D*). Overall, the potency of THZ1 in suppressing tumor growth in our ovarian tumor models is striking, given that tumor regression is rarely observed in previous studies using THZ1. The combination study indicated that combining THZ1 with clinical PARP inhibitors could be promising future therapeutic approach for treating ovarian cancer.

In summary, our study demonstrates an exceptional antitumor activity of THZ1 in models of ovarian cancer and point to at least part of its mechanistic actions whereby an oncogenic transcription factor with widespread alteration in ovarian cancer, MYC, is strongly suppressed by this compound. Thus the current study provides a compelling rationale for investigating THZ1, or its derivatives, for treating MYC-dependent ovarian cancer in the clinic. In addition, we find that suppression of MYC expression can only be achieved by simultaneous inhibition of CDK7, CDK12 and CDK13, further demonstrating the advantage of polypharmacology in overcoming functional redundancy in transcriptional regulation through targeting multiple CDKs.

## Materials and methods

| Reagent type (species) or resource | Designation | Source or reference | Identifiers | Additional information |
|---|---|---|---|---|
| Antibody | Rabbit monoclonal anti-c-MYC (Y69) | ABCAM | Cat# ab32072; RRID: AB_731658 | 1:1000 dilution |
| Antibody | Rabbit polyclonal anti-MCL-1 | Santa Cruz Biotechnology | Cat# sc-819; RRID: AB_2144105 | 1:1000 dilution |
| Antibody | Mouse monoclonal anti-alpha-Tubulin | Cell Signaling Technology | Cat# 3873S; RRID: AB_1904178 | 1:5000 dilution |
| Antibody | Rabbit monoclonal anti-HA tag | Cell Signaling Technology | Cat# 3724; RRID: AB_1549585 | 1:1000 dilution |
| Antibody | Rabbit polyclonal anti-GAPDH | Cell Signaling Technology | Cat# 2118; RRID: AB_561053 | 1:3000 dilution |
| Antibody | Rabbit polyclonal anti-RNA Polymerase II RPB1 | Bethyl Laboratories | Cat# A300-653A; RRID: 519334 | 1:500 dilution |
| Antibody | Rat monoclonal anti-RNA Polymerase II subunit B1 (phosphor CTD ser-2) | EMD Millipore | Cat# 04–1571; RRID: AB_2687450 | 1:1000 dilution |
| Antibody | Rat monoclonal anti-RNA Polymerase II subunit B1 (phosphor CTD ser-5) | EMD Millipore | Cat# 04–1572; RRID: AB_ 2687451 | 1:5000 dilution |

*Continued on next page*

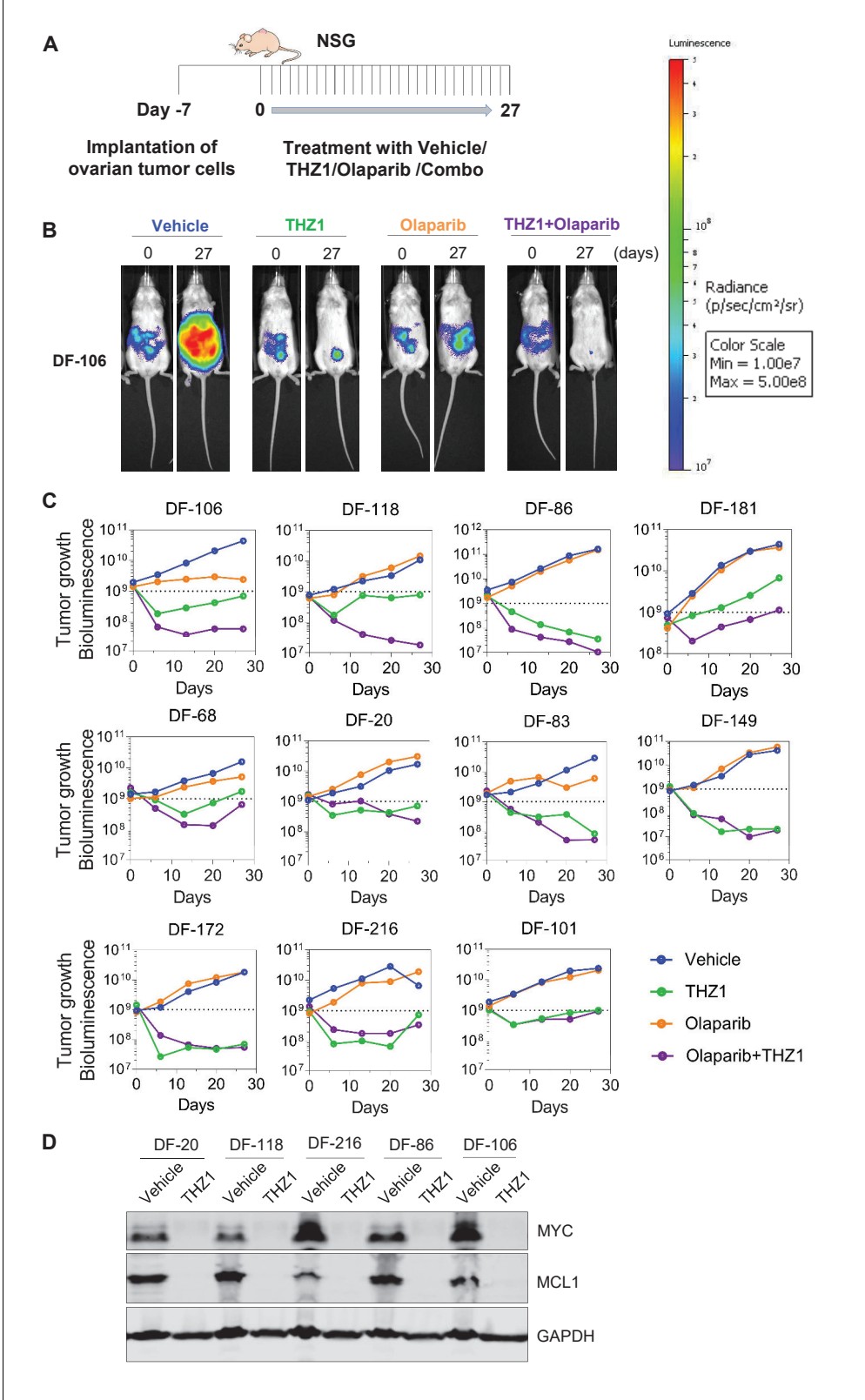

**Figure 5.** THZ1 abrogates the growth of patient-derived high-grade serous ovarian cancer cells in vivo. **(A)** A schematic diagram of the experimental design. **(B)** Luminescence picture of luciferized PDX mice (eg. DF-106) at day 0 and day 27, treated with vehicle control (10 ml/kg, PO, QD) or THZ1 (10 mg/kg, IP, BID) or Olaparib (100 mg/kg, PO, QD) or Combo (THZ1 +Olaparib)., Luminescence scale bar was shown on the right. **(C)** Luminescence
*Figure 5 continued on next page*

*Figure 5 continued*

signal of the tumor in 11 PDX models (n = 44 mice) treated with vehicle control or THZ1 or Olaparib or Combo. (D) Following the last treatment on day 27, tumor ascites from 5 PDX models (n = 10 mice) were harvested for the preparation of whole cell lysates followed by immunoblotting using the indicated antibodies. Also see *Figure 5— figure supplement 1*.

DOI: https://doi.org/10.7554/eLife.39030.013

The following figure supplement is available for figure 5:

**Figure supplement 1.** Body weight measurement of the 11 PDX models (n = 44 mice) treated with vehicle control (10 ml/kg, PO, QD) or THZ1 (10 mg/kg, IP, BID) or Olaparib (100 mg/kg, PO, QD) or Combo.

DOI: https://doi.org/10.7554/eLife.39030.014

*Continued*

| Reagent type (species) or resource | Designation | Source or reference | Identifiers | Additional information |
|---|---|---|---|---|
| Antibody | Rat monoclonal anti-RNA Polymerase II subunit B1 (phosphor CTD ser-7) | EMD Millipore | Cat# 04–1570; RRID: AB_2687452 | 1:5000 dilution |
| Peptide, recombinant protein | THZ1 | Synthesized in our laboratory | | |
| Peptide, recombinant protein | THZ531 | Synthesized in our laboratory | | |
| Peptide, recombinant protein | YKL-1–116 | Synthesized in our laboratory | | |
| Commercial assay or kit | CellTiter-Glo Luminescent Cell Viability Assay | Promega | Cat# G7573 | |
| Commercial assay or kit | RNeasy Plus Mini kit | Qiagen | Cat# 74136 | |
| Commercial assay or kit | High-Capacity RNA-to-cDNA Kit | Applied Biosystems | Cat# 4387406 | |
| Commercial assay or kit | SYBR Select Master Mix | Applied Biosystems | Cat# 4472908 | |
| Cell line (*Homo sapiens*) | KURAMOCHI | Panagiotis A. Konstantinopoulos's laboratory | RRID:CVCL 1345 | |
| Cell line (*Homo sapiens*) | COV362 | Panagiotis A. Konstantinopoulos's laboratory | RRID:CVCL_2420 | |
| Cell line (*Homo sapiens*) | OVCAR8 | Panagiotis A. Konstantinopoulos's laboratory | RRID:CVCL_1629 | |
| Cell line (*Homo sapiens*) | IGROV1 | Panagiotis A. Konstantinopoulos's laboratory | RRID:CVCL_1304 | |
| Cell line (*Homo sapiens*) | SKOV3 | Panagiotis A. Konstantinopoulos's laboratory | RRID:CVCL_0532 | |
| Cell line (*Homo sapiens*) | CAOV3 | Panagiotis A. Konstantinopoulos's laboratory | RRID:CVCL_0201 | |
| Cell line (*Homo sapiens*) | OVCAR3 | Panagiotis A. Konstantinopoulos's laboratory | RRID:CVCL_0465 | |

*Continued*

| Reagent type (species) or resource | Designation | Source or reference | Identifiers | Additional information |
|---|---|---|---|---|
| Cell line (*Homo sapiens*) | OVCAR4 | Panagiotis A. Konstantinopoulos's laboratory | RRID:CVCL_1627 | |
| Cell line (*Homo sapiens*) | OVSAHO | Panagiotis A. Konstantinopoulos's laboratory | RRID:CVCL_3114 | |
| Cell line (*Homo sapiens*) | PEO1 | Panagiotis A. Konstantinopoulos's laboratory | RRID:CVCL_Y032 | |
| Cell line (*Homo sapiens*) | JHOS2 | Panagiotis A. Konstantinopoulos's laboratory | RRID:CVCL_4647 | |
| Cell line (*Homo sapiens*) | 293T | Jean Zhao's laboratory | | |
| Cell line (*Homo sapiens*) | MM1.S | James Bradner's laboratory | | |
| Recombinant DNA reagent | pLenti_crispr_sgGFP | This paper | | Plasmid |
| Recombinant DNA reagent | pLenti_crispr_sgMYC_1 | This paper | | Plasmid |
| Recombinant DNA reagent | pLenti_crispr_sgMYC_2 | This paper | | Plasmid |
| Recombinant DNA reagent | pLenti_crispr_sgMCL1_1 | This paper | | Plasmid |
| Recombinant DNA reagent | pLenti_crispr_sgMCL1_2 | This paper | | Plasmid |
| Recombinant DNA reagent | pTrex_HA_CDK7 WT | This paper | | Plasmid |
| Recombinant DNA reagent | pTrex_HA_CDK7 C312S | This paper | | Plasmid |
| Software, algorithm | GraphPad Prism | Graphpad Software Inc | https://www.graphpad.com/scientific-software/prism/ | |
| Software, algorithm | Image Studio Lite | LI-COR Biosciences | https://www.licor.com/bio/products/software/image_studio_lite/ | |
| Software, algorithm | ImageJ | National Institutes of Health | https://imagej.nih.gov/ij/ | |
| Software, algorithm | Vector NTI | Invitrogen | https://www.thermofisher.com/us/en/home/life-science/cloning/vector-nti-software.html | |

## Cell culture

All cells were grown in RPMI1640 or DMEM medium (Life Technologies), supplemented with 10% fetal bovine serum (Gibco), 50 units/mL penicillin, 50 units/mL streptomycin, and maintained in humidified 37°C/5%$CO_2$ incubator. Ovarian cancer cell lines, including KURAMOCHI, COV362, IGROV1, OVCAR8, OVSAHO, SKOV3, CAOV3, JHOS2, PEO1, OVCAR3 and OVCAR4, were generous gifts from Panos Konstantinopoulos's laboratory at Dana-Farber Cancer Institute. All the cell lines have been tested to be mycoplasma-free using MycoAlert Mycoplasma Detection Kit (Lonza).

## Immunoblotting

Cells were washed once with 1x phosphate buffered saline (PBS) and then lysed in RIPA buffer (50 mM Tris, pH 7.5, 150 mM NaCl, 1% NP-40, 0.5% sodium deoxycholate, and 0.1% SDS) supplemented with protease and phosphatase inhibitors (Roche). Protein concentrations were determined by using the Pierce BCA protein assay kit (Life Technologies). Equal amount of protein was resolved on SDS-PAGE and was subsequently transferred onto nitrocellulose membrane (Bio-Rad). The membrane was blocked with Odyssey block buffer TBS (LI-COR Biosciences) and was then incubated with primary antibodies in 20% of Odyssey block buffer TBST (with 0.1%Tween20) overnight at 4°C with gentle rotating. After washing, the membrane was incubated with fluorophore-conjugated secondary antibodies (1: 10,000) in 20% of Odyssey block buffer TBST (with 0.1% Tween20) for 1 hr at room temperature. The membrane was then washed three times in TBST and scanned with an Odyssey Infrared scanner (Li-Cor Biosciences). Primary antibodies include anti-alpha-Tubulin (Cell Signaling Technology # 3873S), anti-c-MYC (Y69) (ABCAM # ab32072), anti-MCL-1(S-19) (Santa Cruz # sc-819), anti-RNA polymerase II (Bethyl Laboratories #A300-653A), anti-RNA polymerase II subunit B1 (phospho CTD Ser-2, clone 3E10) (Millipore #04–1571), anti-RNA polymerase II subunit B1 (phospho-CTD Ser-5, clone 3E8) (Millipore #04–1572), anti-RNA polymerase II subunit B1 (phospho-CTD Ser-7, clone 4E12) (Millipore #04–1570), GAPDH (Cell Signaling Technology #2118), and anti-HA-tag (C29F4) (Cell Signaling Technology #3724). Secondary antibodies include IRDye 800CW Goat anti-Rabbit IgG (LICOR Biosciences #926–32211), IRDye 680LT Goat anti-Mouse IgG (LICOR Biosciences #926–68020), and IRDye 680RD Goat anti-Rat IgG (LICOR Biosciences #925–68076).

## Plasmids and primers

Plasmids: pLenti_CRISPR_v2 vector was ordered from Addgene (Addgene #52961).

pLenti_sgGFP, pLenti_sgMYC_1, pLenti_sgMYC_2, pLenti_sgMCL1_1, pLenti_sgMCL1_2, pTrex_-HA_cdk7 WT, and pTrex_HA_cdk7 C312S lentiviral plasmids were made in our laboratory.

Primers for cloning sgRNA into pLenti_CRISPR_v2 vector
sgGFP_1F: CACCGGGGCGAGGAGCTGTTCACCG
sgGFP_1R: AAACCGGTGAACAGCTCCTCGCCCC
sgMYC_1F: CACCGAACGTTGAGGGGCATCGTCG
sgMYC_1R: AAACCGACGATGCCCCTCAACGTTC
sgMYC_2F: CACCGGCCGTATTTCTACTGCGACG
sgMYC_2R: AAACCGTCGCAGTAGAAATACGGCC
sgMCL1_1F: CACCGGCTTCCGCCAATCACCGCGC
sgMCL1_1R: AAACGCGCGGTGATTGGCGGAAGCC
sgMCL1_2F: CACCGCTCGGCCCGGCGAGAGATAG
sgMCL1_2R: AAACCTATCTCTCGCCGGGCCGAGC
Primers for qPCR:
GAPDH_qPCR_forward: GGTCTCCTCTGACTTCAACA
GAPDH_qPCR_reverse: GTGAGGGTCTCTCTCTTCCT
MYC_qPCR_forward: GGCTCCTGGCAAAAGGTCA
MYC_qPCR_reverse: CTGCGTAGTTGTGCTGATGT
MCL1_qPCR_forward: TGCTTCGGAAACTGGACATCA
MCL1_qPCR_reverse: TAGCCACAAAGGCACCAAAAG

## Virus infection

Lentiviruses were generated in HEK293T cells by transfecting cells with packaging DNA plus pLenti_-CRISPR plasmids. Typically, 2 µg pLenti_CRISPR plasmid encoding sgRNA, 1.5 µg pCMVdR8.91, and 0.5 µg pMD2-VSVG, 12 µl Lipofectamin2000 (Invitrogen) were used. DNA and lipid were pre-diluted in 300 µl Opti-MEM (Invitrogen) individually and then mixed well gently. After 30 min of incubation at RT, the DNA-lipid mixtures were added dropwise to HEK293T cells (2 × $10^6$ cells were seeded in one T-25 flask, one day prior to transfection). Viral supernatant was collected 2 and 3 days after transfection, filtered through 0.45 µm membranes, and added to target cells in the presence of polybrene (8 µg/ml, Millipore). Target cells were infected twice with the virus at 48 hr and 72 hr later. Puromycin (1 µg/ml for KURAMOCHI and OVCAR8; 2 µg/ml for SKOV3) was used to treat cells for 2 days for selection, which eliminated all cells in an uninfected control group. Cells were

harvested 4 days after the initial viral infection and subjected them for either western blotting to assess the knockdown efficiency or for clonogenic cell growth assay. For lentiviral infection with pTrex_HA_cdk7 WT and pTrex_HA_cdk7 C312S plasmids, cells were infected as described above, and were selected with G418 (1 mg/ml) for 5 days then treated with 0.2 µg/ml Doxycycline for 1 day to induce the HA tagged-CDK7 overexpression. Doxycycline was replenished every 3 days during the following assays.

## Cell proliferation assays

After virus infection and selection with puromycin, cells were seeded in 12-well plates (at the density of $5 \times 10^3$) in 1 ml medium. 14 days later, cells were fixed with 1% formaldehyde for 15 minutes, and stained with crystal violet (0.05%, wt/vol), a chromatin-binding cytochemical stain for 15 minutes. The plates were washed extensively in plenty of deionized water, dried upside-down on filter paper, and imaged with Epson scanner.

For the 3-day cell proliferation assay in 96-well plate, cells were plated at the density of 6000 to 10,000 cells per well and treated with THZ1 or YKL-1–116 of various concentrations on the next day. After 72 hr incubation, CellTiter-Glo reagent (Promega #G7572) was added to cells directly and luminescent signal was read on a plate reader (Perkin Elmer EnVision).

## RNA extraction

Cells were plated in six-well plates and allowed to adhere overnight. Following treatment with compounds for 6 hr, cells were lysed, homogenized using QIAshredder spin column (Qiagen #79654), and subjected to total RNA extraction using RNeasy Plus Mini kit (Qiagen #74136). This kit contains gDNA columns to remove genomic DNA, according to the manufacturer's instructions.

## RT-qPCR

First strand cDNA was synthesized from 2 µg of total RNA using The High Capacity cDNA Reverse Transcription Kit (Life technologies #4368814). The cDNAs were diluted 15-fold in deionized water and then mixed with specific primers (10 µM) and 2X SYBR Select Master Mix (Applied Biosystems #4472908). The reactions were set up in MicroAmp Fast Optical 96-Well Reaction Plate (Life Technologies #4346906) and were run on 7500 Real-Time PCR System (Applied Biosystems). The *GAPDH* gene was used as a housekeeping gene control. Relative gene expression was calculated using the comparative method ($2–\Delta\Delta Ct$).

## RNA-sequencing

Following total RNA extraction, cell-count-normalized RNA samples were mixed with the RNA standards - External RNA Control Consortium (ERCC) Spike-In Mix (Ambion, 4456740) prior to library construction (*Lovén et al., 2012*). Libraries were prepared using TruSeq Stranded mRNA Library Prep Kit (Illumina), and equimolar libraries were multiplexed and sequenced on an Illumina NextSeq 500 (single end 75 bp reads) by the Molecular Biology Core Facility at the Dana-Farber Cancer Institute. Fastq files were aligned to human genome build hg19 using HiSat with default parameters. Transcripts were assembled, and Fragments Per Kilobase of transcript per Million mapped reads (FPKM) values were generated using cuffquant and cuffnorm from the cufflinks pipeline (*Trapnell et al., 2010*). FPKM values were then normalized to synthetic ERCC spike-in RNAs as described previously (*Lovén et al., 2012*). A transcript was considered to be expressed in each data set if in at least one experimental condition the normalized FPKM >1.

## Animal studies

All animal experiments were conducted in accordance with the animal use guidelines from the NIH and with protocols (Protocol # 11–044) approved by the Dana-Farber Cancer Institute Animal Care and Use Committee. We used the ovarian patient-derived xenografts (PDX) models that we established previously (*Liu et al., 2017*). The primary tumor cells were transduced with luciferase gene to enable the use of non-invasive bioluminescent imaging (BLI) for measurement of tumor growth. Briefly, ovarian cancer cells were taken from consented patients with HGSOC and implanted intra-peritoneally into immunocompromised NOD-SCID IL2Rγnull mice (NSG, Jackson Laboratory). $5 \times 10^6$ ascites-derived cells were implanted in each mouse and 7 days post implantation, mice were

imaged by BLI and assigned to four groups of treatment with vehicle control via oral gavage (PO) once daily (QD) at the dose of 10 ml/kg; THZ1 intraperitoneally (IP) twice daily (BID) at the dose of 10 mg/kg; Olaparib via PO, QD at the dose of 100 mg/kg; the combination of THZ1 and Olaparib. Tumor growth was assessed every 7 days (0, 6, 13, 20, 27 days) using BLI until day 27. Upon harvesting, tumors or other tissues were snap-frozen in liquid nitrogen for preparation of lysates and immunoblotting.

### Data availability
RNA-sequencing data reported in this paper has been deposited to the NCBI GEO and are available under the accession number GSE116282.

## Acknowledgements

We thank Dr. Yubao Wang at Dana-Farber Cancer Institute for fruitful discussions of the study and his critical reading of the manuscript. The study was supported by NIH R01 CA197336-02 (NSG), R01 CA179483-02 (NSG), Department of Defense W81XWH-14-OCRP-OCACAOC140632 award (PAK), Cancer Prevention Research Institute of Texas RR150093 (CYL), NIH and NCI R01CA215452-01 (CYL), Pew-Stewart Scholar for Cancer Research (CYL), and American Cancer Society Postdoctoral Fellowship PF-17-010-01-CDD (BN).

## Additional information

### Competing interests
Nicholas P Kwiatkowski, Tinghu Zhang: is an inventor on a patent application covering THZ1 (patent application number WO/2014/063068 A1), which is licensed to Syros Pharmaceuticals. Charles Y Lin: is a consultant of Jnana Therapeutics and is a shareholder and inventor of IP licensed to Syros Pharmaceuticals. Nathanael S Gray: is an inventor on a patent application covering THZ1 (patent application number WO/2014/063068 A1), which is licensed to Syros Pharmaceuticals. Is a scientific founder and equity holder of Syros Pharmaceuticals, C4 Therapeutics, Petra Pharma, Gatekeeper Pharmaceuticals, and Soltego. The other authors declare that no competing interests exist.

### Funding

| Funder | Grant reference number | Author |
|---|---|---|
| National Cancer Institute | NIH R01 CA197336-02 | Nathanael S Gray |
| National Cancer Institute | NIH R01 CA179483-02 | Nathanael S Gray |
| U.S. Department of Defense | W81XWH-14-OCRP-OCACAOC140632 award | Panagiotis A Konstantinopoulos |
| Cancer Prevention and Research Institute of Texas | RR150093 | Charles Y Lin |
| National Cancer Institute | R01CA215452-01 | Charles Y Lin |
| American Cancer Society | Postdoctoral Fellowship PF-17-010-01-CDD | Behnam Nabet |

The funders had no role in study design, data collection and interpretation, or the decision to submit the work for publication.

### Author contributions
Mei Zeng, Conceptualization, Data curation, Formal analysis, Validation, Investigation, Visualization, Writing—original draft, Writing—review and editing; Nicholas P Kwiatkowski, Conceptualization, Data curation, Supervision, Project administration, Writing—review and editing; Tinghu Zhang, Resources, Supervision, Project administration; Behnam Nabet, Data curation, Software, Methodology; Mousheng Xu, Data curation, Software, Formal analysis, Methodology; Yanke Liang, Chunshan Quan, Mingfeng Hao, Alan L Leggett, Investigation, Methodology; Jinhua Wang, Khyati Meghani, Resources, Investigation; Sangeetha Palakurthi, Data curation, Supervision, Investigation; Shan Zhou, Qing Zeng, Data curation, Investigation; Paul T Kirschmeier, Supervision, Investigation; Jun Qi,

Resources, Funding acquisition; Geoffrey I Shapiro, Joyce F Liu, Resources, Supervision; Ursula A Matulonis, Resources, Formal analysis, Funding acquisition; Charles Y Lin, Data curation, Formal analysis, Validation, Investigation, Methodology, Writing—review and editing; Panagiotis A Konstantinopoulos, Resources, Supervision, Writing—original draft, Project administration, Writing—review and editing; Nathanael S Gray, Conceptualization, Resources, Supervision, Funding acquisition, Project administration, Writing—review and editing

### Author ORCIDs

Mei Zeng ⓘ http://orcid.org/0000-0001-5316-3522
Behnam Nabet ⓘ https://orcid.org/0000-0002-6738-4200
Charles Y Lin ⓘ https://orcid.org/0000-0002-9155-090X
Nathanael S Gray ⓘ http://orcid.org/0000-0001-5354-7403

### Ethics

Animal experimentation: All animal experiments were conducted in accordance with the animal use guidelines from the NIH and with protocols (Protocol # 11-044) approved by the Dana-Farber Cancer Institute Animal Care and Use Committee. Full details are described in Materials and Methods - Animal Studies.

### Decision letter and Author response

Decision letter https://doi.org/10.7554/eLife.39030.022
Author response https://doi.org/10.7554/eLife.39030.023

## Additional files

### Supplementary files

• Supplementary file 1. Table of epigenetic and transcriptional compound screen.
DOI: https://doi.org/10.7554/eLife.39030.016

• Supplementary file 2. Table of normalized MYC expression values.
DOI: https://doi.org/10.7554/eLife.39030.017

• Transparent reporting form
DOI: https://doi.org/10.7554/eLife.39030.018

### Data availability

RNA sequencing data have been deposited in GEO under accession code GSE116282.

The following dataset was generated:

| Author(s) | Year | Dataset title | Dataset URL | Database and Identifier |
|---|---|---|---|---|
| Zeng M, Xu M, Charles Y Lin, Nathanael S Gray | 2018 | Targeting MYC dependency in ovarian cancer through inhibition of CDK7 and CDK12/13 | https://www.ncbi.nlm.nih.gov/geo/query/acc.cgi?acc=GSE116282 | Gene Expression Omnibus, GSE116282 |

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
