## [Decision Letter]

Thank you for submitting your article "Targeting MYC dependency in ovarian cancer through inhibition of CDK7 and CDK12/13" for consideration by *eLife*. Your article has been reviewed by three peer reviewers, one of whom is a member of our Board of Reviewing Editors, and the evaluation has been overseen by Charles Sawyers as the Senior Editor. The reviewers have opted to remain anonymous.

The reviewers have discussed the reviews with one another and the Reviewing Editor has drafted this decision to help you prepare a revised submission.

Summary:

This is a relatively straightforward study that proposes that transcriptional downregulation of MYC and possibly MCL1 through inhibition of CDKs7, 12 and 13 is a potential therapeutic avenue for MYC amplified or overexpressed ovarian cancer. While the authors have conducted a nice series of experiments, the conclusions are overly biased towards effects on MYC and MCL1 and disregard other transcriptional effects of THZ1 that are evident from the study. Moreover, the authors must do a better job of acknowledging the potential contribution of off-targets (i.e. proteins other than CDK7, CDK12, or CDK13) in this study and they need further experiments to verify whether CDK7 and/or CDK12/13 are in fact the relevant targets. These issues are particularly important given many of the conclusions have been made in previous papers, with a difference here being a focus on ovarian cancer.

Essential revisions:

1) The lack of clear mechanistic insight into how THZ1 inhibits MYC expression and target expression in MYC amplified tumors is a relative shortcoming. How do the authors believe that this agent can reduce the expression of a gene which is on a recurrent amplicon? Are there other genes on 8q whose expression is increased and how are they affected by the drug? Specifically, the data presented clearly shows that *PVT1* is also amplified and increased in its expression, which has been demonstrated to independently contribute to ovarian cancer pathogenesis. Is *PVT1* a transcriptional target of THZ1/JQ1 in these cells? In addition, CDK7 inhibition by YKL-1-116 alone was able to decrease mRNA levels of MYC, whereas THZ-531 (targeting CDK12 and CDK13) was not. This effect appears to be cell type-specific, because the results in Figure 3 (KURAMOCHI line) were different and somewhat more consistent with the statement in the Abstract. Pursuant to this, the authors have not conclusively demonstrated that CDK7, CDK12, or CDK13 are the relevant targets in this study. At minimum, the authors should test point mutants of CDK7 and/or CDK12 and CDK13 to confirm that they are the relevant targets in these cell lines.

2) The relative impact of the drug on MYC and MCL-1 vs. downstream targets is not clear. Can this be delineated further? Genetic rescue of MYC and MCL1 should also be performed to provide more than correlative data. Moreover, the effects observed of the MYC targeting sgRNAs is attributed to the loss of MYC expression, however, a major problem with targeting sgRNAs to amplified regions in the genome is the induction of DNA damage and cell death, independent of the sgRNA target (Aguirre AJ et al., 2017, Cancer Discovery). Are these cells responsive to shRNA mediated depletion of MYC or can the effect be rescued by MYC overexpression?

3) The PDX data is the most exciting. One wonders if there are agents used in R/R ovarian cancer which can be tested in combination to determine if the first studies should be monotherapy or combination studies.

---

## [Author Response]

Essential revisions:1) The lack of clear mechanistic insight into how THZ1 inhibits MYC expression and target expression in MYC amplified tumors is a relative shortcoming. How do the authors believe that this agent can reduce the expression of a gene which is on a recurrent amplicon? Are there other genes on 8q whose expression is increased and how are they affected by the drug? Specifically, the data presented clearly shows that PVT1 is also amplified and increased in its expression, which has been demonstrated to independently contribute to ovarian cancer pathogenesis. Is PVT1 a transcriptional target of THZ1/JQ1 in these cells?

This is an important question. Our study includes several lines of evidence that CDK7 inhibitors produce highly potent and on-targeting inhibition in suppressing MYC expression. For example, treating a variety of ovarian cancer cell lines with THZ1 for brief periods (4 – 6 hours) at nanomolar concentrations (50-250 nM) can nearly abolish the protein expression of MYC, as well as its transcript (as measured by qPCR or RNA-seq) (Figure 2C-D; Figure 3B). We further demonstrate that the robust effects on MYC expression can be fully rescued by expressing THZ1-resistant form of CDK7 (see below response to point #2). All these observations strongly suggest that CDK7 inhibitors can suppress MYC expression in MYC-amplified ovarian cancer cells. The reviewer poses an intriguing question regarding whether amplified genes are subject to transcriptional inhibition. Although such a topic has not been extensively studied, we know from literature that a number of amplified oncogenes are susceptible to transcriptional inhibition, such as targeting MYCN in neuroblastoma cells (Chipumuro et al., 2014). Similarly, BET inhibitors were found to suppress MYC expression in leukemia cell lines with MYC amplification (Mertz et al., 2011).

The regulation of MYC expression by THZ1 appears quite specific, because *PVT1* − a lncRNA gene co-amplified with MYC on 8q24 − is not significantly downregulated by THZ1 (see Figure 3—figure supplement 1).

In addition, CDK7 inhibition by YKL-1-116 alone was able to decrease mRNA levels of MYC, whereas THZ-531 (targeting CDK12 and CDK13) was not. This effect appears to be cell type-specific, because the results in Figure 3 (KURAMOCHI line) were different and somewhat more consistent with the statement in the Abstract.

Please see Author response image 1 (adapted from manuscript Figure 4B; Figure 4—figure supplement 1A). DK7 inhibition by a structurally distinct CDK7 inhibitor, YKL-1-116, was able to decrease both MYC mRNA level and protein abundance, and this effect was consistently observed in both KURAMOCHI and COV362 cells (Author response image 1).

The CDK12/13 inhibitor THZ531 exhibits quite unique effects on MYC expression. At low doses of THZ531, MYC expression is induced while higher doses of THZ531 cause MYC repression (Author response image 1). The kinetics of inhibition of MYC expression by THZ531 is subtly different among these cell lines: MYC expression was induced by THZ531 at low dosage (0.05 or 0.1 μM) and then suppressed by higher concentrations (0.5 μM and above) in KURAMOCHI cells; while in COV362 cells, MYC expression was induced by THZ531 at 0.5 μM and above but reduced at 5 μM. Therefore, we think that THZ531 impacts MYC expression with a similar pattern in different cell lines, although with subtle differences in the required compound dosages.

**Author response image 1. respfig1:** CDK7 inhibitor (YKL-1-116) and CDK12/13 inhibitor (THZ531) have different effects on MYC expression. Indicated ovarian cancer cell lines were treated with increasing concentrations of YKL- 1-116 or THZ531, with cell lysates prepared following 6 hours of treatment.

Pursuant to this, the authors have not conclusively demonstrated that CDK7, CDK12, or CDK13 are the relevant targets in this study. At minimum, the authors should test point mutants of CDK7 and/or CDK12 and CDK13 to confirm that they are the relevant targets in these cell lines.

We agree that the raised point is extremely critical. To further validate that CDK7 is the relevant target of THZ1 or YKL-1-116, we now ectopically overexpress a mutant form of CDK7 (C312S) that cannot be covalently bound by the inhibitors and thus confer resistance to the compounds (Kwiatkowski et al., 2014). We find that the mutant CDK7 (C312S), but not the wild type, effectively rescues THZ1-induced MYC downregulation (Figure 4—figure supplement 2A). Concurrently, other readouts of CDK7 activity, such as CTD phosphorylation of RNAPII at Ser 5 and Ser 7, are also rescued by this CDK7 mutant. These data indicate that CDK7 is indeed a target of THZ1 in downregulating MYC.

Although overexpression of mutant CDK7 (C312S) fails to rescue cell growth inhibition conferred by THZ1 (see Author response image 2), it significantly rescues cell growth inhibition by the selective CDK7 inhibitor, YKL-1-116 (Author response image 2). We would expect that a greater degree of rescue might require the elimination of endogenous CDK7. In addition, considering that THZ1 targets both CDK7 and CDK12/13, we speculate that co-overexpression of mutant CDK7 (C312S), mutant CDK12 (C1039S), and mutant CDK13 may rescue cell growth inhibition by THZ1. Unfortunately, we failed to successfully achieve overexpression of CDK12/13, likely due to their large sizes (1490 aa for CDK12, and 1512 aa for CDK13).

**Author response image 2. respfig2:** Overexpressing CDK7 C312S mutant rescued cell growth inhibition by YKL-1-116.

OVCAR8 cells were infected with or without HA-tagged CDK7 WT or C312S, followed by G418 selection and doxycycline induction, and then were treated with increasing concentrations of (A) THZ1, top panel, or (B) YKL- 1-116, bottom panel.

Left. Cells were then subjected to CellTiter-Glow Luminescent Cell Viability Assay after 72 hours of treatment and data were represented as mean ± SD of biological triplicates. Right. Cells were treated with increasing concentrations of THZ1 (0, 50, 100, 200 nM) or of YKL-1-116 (0, 62.5, 125, 250, 500, 1000 nM), and subjected to clonogenic cell growth. Cells were fixed 7 days after treatment and stained with crystal violet.</Author response image 2 title/legend>

2) The relative impact of the drug on MYC and MCL-1 vs. downstream targets is not clear. Can this be delineated further? Genetic rescue of MYC and MCL1 should also be performed to provide more than correlative data.

We have found that (1) MYC or MCL-1 is functionally important for ovarian cancer cell growth, and (2) MYC expression is efficiently downregulated by CDK7 inhibition – an observation that we are now able to rescue through expressing the mutant CDK7 (C312S) that is defective in binding to and suppressing CDK7. It is likely that the downregulation of MYC, MCL-1, and others, act collectively to account for the cell growth phenotypes induced by CDK7 inhibition.

Upon the suggestion, we study if ectopic expression of MYC could rescue cell growth inhibition conferred by CDK7 inhibitors. We overexpressed MYC, or MCL-1, or MYC plus MCL-1 in ovarian cancer cells by lentiviral infection (see Author response image 3).

**Author response image 3. respfig3:** Overexpression of GFP/ MYC/ MYC-1/ MYC+MCL-1 in OVCAR8 cells.

Cells were infected with pCDH-GFP, or MYC, or MCL-1, or MYC plus MCL-1 twice, followed by selection with puromycin (1 ug/ml, 2 days). Cells were then seeded at 0.2 million per well in a 12-well plate, then treated with increasing concentrations of THZ1 (0, 31.25, 62.5, 125, 250, 500 nM) the next day, with cell lysates prepared following 6 hours of treatment.</Author response image 3 title/legend>

Next, we examined whether overexpressing MYC, or MCL-1, or MYC plus MCL-1 could rescue cell growth inhibition observed upon treatment with THZ1 or YKL-1-116. This has turned out to be technically difficult, because we found that overexpressing MYC inhibits the growth of ovarian cancer cells (cells that already have high endogenous level of MYC expression; see Author response image 4). The observation indicates that too much MYC protein – such as by overexpressing MYC in cells with MYC-amplification – could be toxic. Notably, similar observations have been documented in a number of models (Bissonnette et al., 1992; Muthalagu et al., 2014; Topham et al., 2015). We therefore have to abandon this line of experimentation.

**Author response image 4. respfig4:** Overexpression of MYC inhibits cancer cell growth. OVCAR8 cells were infected with pCDH-GFP/ MYC/ MYC-1/ MYC+MCL-1 twice, then selected with puromycin 1 ug/ml for 2 days. Cells were seeded at 5,000 per well in a 12-well plate, then treated with DMSO or increasing concentrations of THZ1 or YKL-1-116 the next day. Pictures were taken for cells treated with DMSO control for 6 days prior to cell fixation.

Moreover, the effects observed of the MYC targeting sgRNAs is attributed to the loss of MYC expression, however, a major problem with targeting sgRNAs to amplified regions in the genome is the induction of DNA damage and cell death, independent of the sgRNA target (Aguirre AJ et al., 2017 Cancer Discovery). Are these cells responsive to shRNA mediated depletion of MYC or can the effect be rescued by MYC overexpression?

This is an excellent question. We now analyzed the large-scale CRISPR screen performed by Broad Institute, a study where they developed the CERES computational model to reduce the false-positive differential dependencies caused by multiple DNA breaks resulted from targeting amplified regions (Meyers et al., 2017). In the analysis illustrating MYC dependency in a total 484 cancer cell lines, there is no statistical correlation between MYC dependency values and with MYC copy number (see Figure 1—figure supplement 1A), indicating the successful computational elimination of effects introduced by targeting amplified MYC. Notably, ovarian cancer cells overall demonstrate a high dependence on MYC (indicated by the low CERES values), and as expected, MYC dependency is highly correlated with cell dependency on MAX – a partner of MYC for transcriptional regulation (Figure 1—figure supplement 1B). These data further confirm the functional roles of MYC for ovarian cancer cell proliferation.

3) The PDX data is the most exciting. One wonders if there are agents used in R/R ovarian cancer which can be tested in combination to determine if the first studies should be monotherapy or combination studies.

We performed a combination study with THZ1 and PARP inhibitor Olaparib, a FDA-approved drug in relapsed ovarian cancer irrespective of BRCA1/2 status. We have now included the data in the revised manuscript (see Figure 5A-C). Although the single-agent efficacy of Olaparib is quite modest, PARP inhibition enhances the tumor growth inhibition by THZ1 in 5 out of 11 PDX models (DF-106, 118, 86, 68, and 181).